

# Adaption of an array spectroradiometer for total ozone column retrieval using direct solar irradiance measurements in the UV spectral range

Ralf Zuber[1], Peter Sperfeld[2], Stefan Riechelmann[2], Saulius Nevas[2], Meelis Sildoja[2], Gunther Seckmeyer[3]

[1]Gigahertz-Optik GmbH, 82299 Türkenfeld/Munich, Germany
[2]Physikalisch-Technische Bundesanstalt (PTB), Bundesallee 100, 38116 Braunschweig, Germany
[3]Leibniz Universität Hannover, Institute of Meteorology and Climatology, Germany

*Correspondence to*: Ralf Zuber (r.zuber@gigahertz-optik.de), Peter Sperfeld (peter.sperfeld@ptb.de)

**Abstract.** A compact array spectroradiometer technology that enables precise and robust measurements of UV spectral irradiance is presented. The spectroradiometer design allows various applications such as measurement of high-power UV lamps, risk assessment of sources and measurement of solar irradiance. We show that this technology can perform precise total ozone column (TOC) retrieval. The internal stray light, which is often the limiting factor for measurements in the UV spectral range and increases the uncertainty for TOC analysis significantly, is physically reduced so that no other stray light reduction methods, such as mathematical corrections, are necessary. The instrument and its elaborate components have been extensively characterised at the Physikalisch-Technische Bundesanstalt (PTB) in Germany. During an international total ozone measurement intercomparison at the Izaña observatory in Tenerife, the high-quality applicability of the instrument was verified with measurements of the direct solar irradiance and subsequent TOC evaluations based on the spectral data between 12 and 30 September 2016. The results showed deviations of less than 1.5 % to most other instruments in most situations not exceeding 3 % compared to established TOC measurement systems such as Dobson or Brewer.

## 1. Introduction

Many applications in the ultraviolet spectral range cannot be addressed with array spectroradiometers since they are often limited by internal stray light effects (Egli *et al.*, 2016). An example is an accurate measurement of solar irradiance in the UV-B spectral range. The intense radiation of the sun in the visible (VIS) and infrared (IR) generates stray light within the spectrometer, which often dominates over the less intense solar UV-B radiation. However, an accurate measurement of solar spectrum (Seckmeyer *et al.,* 2001, Seckmeyer *et al.,* 2010) is the basis for an accurate evaluation of many derived quantities such as total ozone column (Dobson, 1931, Mayer *et al*., 1998). Hence, these measurements are often performed with double-monochromator-based systems (Hülsen *et al.,* 2016), offering high stray light reduction capabilities. Operating such instruments is often time- and cost-intensive and requires controlled ambient conditions and highly experienced personnel.

To overcome such limitations, Gigahertz-Optik GmbH developed the BTS2048-UV-S series array spectroradiometer. The system conjoins compact instrument design, a physically filter based stray light correction and versatile radiometric applicability. One of the newly developed devices has been adapted specifically for direct solar irradiance measurements and was extensively characterised at PTB. It then took part at the ATMOZ intercomparison campaign in Izaña, Tenerife in 2016 in the framework of the EMRP project ENV59 ATMOZ – a total ozone measurement intercomparison organised by the Izaña Atmospheric Research Center of the Spanish Meteorological Agency (AEMET) and the World Radiation Center



(PMOD-WRC), where new instruments and techniques developed within the project were compared to well established Dobson and Brewer methods.

## 2. Instrument design

The BTS2048-UV-S series array spectroradiometers are based on the well-known Czerny-Turner (Shafer *et al.*, 1964)
5   spectrometer design. The spectrometer uses a back-thinned CCD detector with 2048 pixels and an electronic shutter integrated in a compact optical bench. Integration times from 2 µs up to 60 s provide a high dynamic range of the instrument in the spectral range from 200 nm to 430 nm. The detector unit is complemented with a SiC photodiode to enable fast time-resolved radiometric measurements.

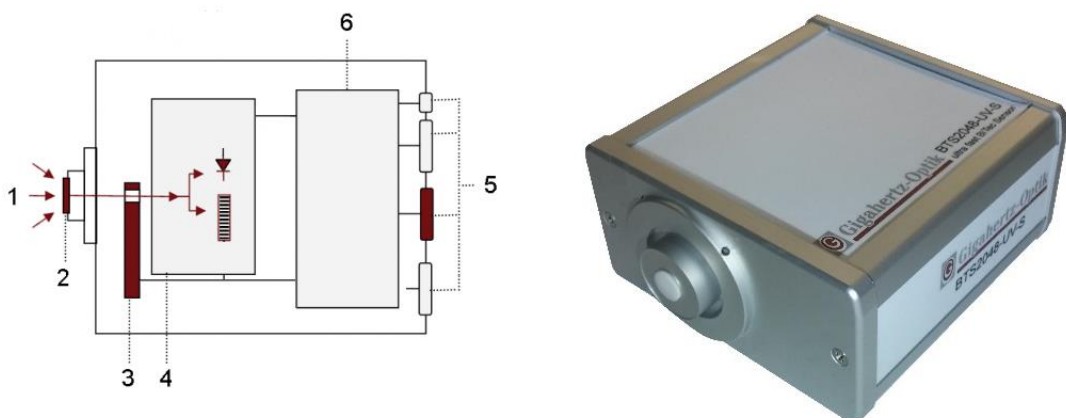

**Figure 1** Schematical setup of the BTS2048-UV-S and photo of the instrument. 1) Incoming optical radiation 2) Direct entrance port with cosine diffuser 3) Filter wheel 4) Sensor system 5) Electrical connectors 6) Microprocessor for data processing and communication.

To enable stray light-corrected measurements, a miniaturised filter wheel with up to six different optical filters is integrated in the optical path between the entrance optic and spectrometer unit (Figure 1). A set of selected optical filters, such as
15   bandpass- and edge-type filters, can be used to preselect the radiation entering the sensor system. In the device one longpass filter and four interference filters with different wavelength ranges are integrated. Hence, several sub-measurements with different filters in the optical path can be performed. The combination of these sub-measurements allows for optimised stray light-reduced spectra. In addition, multiple different combinations of filters, integration times and sub-measurements optimises the measurement scenario to any kind of specific radiometric application. For instance, a so-called out-of-range
20   (OoR) stray light correction method has been implemented, where an additional measurement with a long-pass filter is performed to quantify contribution of the out-of-range stray light to the measured signal, which can then be subtracted. In order to perform reliable solar UV measurements, a specific measurement scenario, the so-called solar bandpass correction method was used. This measurement scenario is based on a series of measurements with several narrow bandpass filters which complement each other (Shaw *et al.,* 2008). Hence, the overall measurement time to get a full spectral measurement is



the sum of all integration times of the sub measurements. Typically this measurement time is in the range of a few seconds, depended on the light source to measure. This active stray light correction process is a straightforward alternative to mathematical stray light correction methods that have been established for array spectroradiometers (Zong *et al.,* 2006, Nevas *et al.,* 2012,). In some cases, e.g. when silicon (Si) detectors are used solely in the UV spectral range, the introduced

technology can reduce the stray light more efficiently. The mathematical stray light correction methods are based on a precise characterisation of the optical imaging performance with so-called line spread functions (LSF). These functions should be determined at each detection wavelength of the spectroradiometer. This so-called in-range stray light can be corrected to improve the measurement threshold by about two orders of magnitude (Zong *et al.,* 2006). However, this correction method is not able to correct for so-called out-of-range stray light that is generated in spectral intervals outside the

spectroradiometer range where the detector used is still sensitive. If, for instance, a Si detector-based spectroradiometer is designed for the spectral range from 200 nm to 400 nm, the in-range stray light can only be corrected for this spectral range. However, the Si detector itself is radiation-sensitive up to 1100 nm and stray light originating from this out-of-range spectral region cannot be characterised with LSF's. This limitation might be resolved by measuring the responsivity of the spectroradiometer to the radiation at OoR wavelength using a calibrated detector or an additional spectroradiometer with

extended wavelength range, for Si ideally up to 1100 nm (Nevas *et al.,* 2014). However, such a correction method requires knowledge about the OoR spectrum not only during instrument calibration, which generally is readily available from the standard lamp calibration data, but also for the radiant sources under investigation, e.g. the direct solar irradiance, which may not be available for practical reasons. Hence, the presented optical filter-based stray light correction technology of the BTS2048-UV-S series offers an attractive alternative, especially in the UV range.

## 3. Characterisation

At PTB, several instrument parameters have been characterised to verify the quality and measurement capability of the BTS2048-UV-S. The wavelength calibration was performed with the help of wavelength tuneable laser systems and checked using mercury pen lamps. The uncertainty for the wavelength calibration was found to be better than 0.1 nm. For measurements of solar irradiance however, the wavelength scale was additionally adapted with standard deviations better

than 0.02 nm to the solar Fraunhofer lines using the MatShic algorithm (Egli *et al.,* 2014). The spectral bandpass was determined using line spread functions (LSF) measured with tuneable laser systems (Nevas *et al.,* 2014). The bandpass function appear to be nearly symmetrical below 360 nm with an average bandwidth of 0.6 nm, full width at half-maximum (FWHM) (see Figure 2).





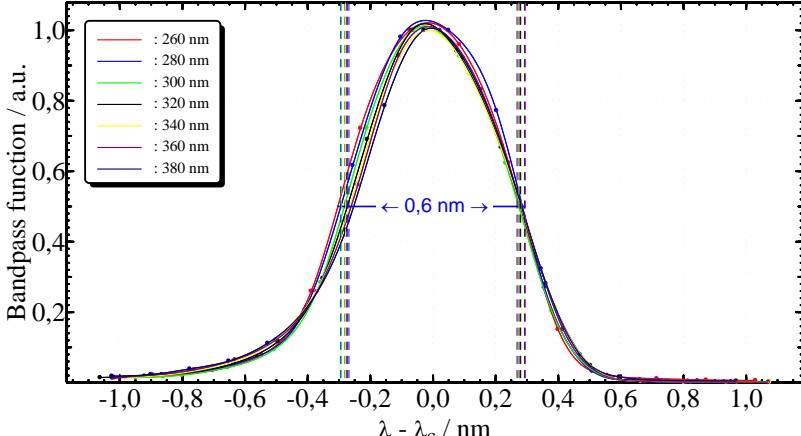

**Figure 2** Bandpass function around centroid $\lambda_c$ for different wavelengths with bandwidth (FWHM, dashed lines) indicated.

The linearity of the BTS2048-UV-S was tested using both the integration time and the irradiance variation methods. For the integration time method, the spectral irradiance of the incident radiation is kept constant while the instrument's integration

5   time is varied over a wide range. Although the measurement signal in raw counts is then varying with the integration time, the normalised count rates per second should remain constant for all measurements. The irradiance variation is performed at constant integration times of the instrument, while the irradiance is linearly reduced between each measurement. Here the ratio of measurement signal (count rate) and spectral irradiance should remain constant for all measurements. By applying a mathematical correction for nonlinearity, the spectrometer showed linearity with a deviation smaller than 1 % over the full

10   dynamic range for the characterised measurement mode.

The instrument has been radiometrically calibrated using 250 W halogen lamps and 30 W deuterium lamps as transfer standards to perform spectral irradiance measurements traceable to the PTB (Sperfeld *et al.,* 2010). Although these standard lamps required a comparably long integration time, due to a high dynamic range of the instrument (typically $5 \cdot 10^{-5}$ W/(m²nm) to $5 \cdot 10^{4}$ W/(m²nm) at 300 nm) it was possible to perform reliable measurements with very short integration

15   times on high-power UV sources, such as medium-pressure mercury lamps, without the need for an additional attenuation.

The used standard lamps allow recalibration of the instrument in the laboratory and in the field. During the measurement campaign, described below, the radiometric calibration of the BTS2048-UV-S could be verified with a standard deviation of less than 1 % compared to the calibrations in the laboratory before and after the campaign.



## 4. Performance evaluation

### 4.1 Intercomparison of spectroradiometric measurements

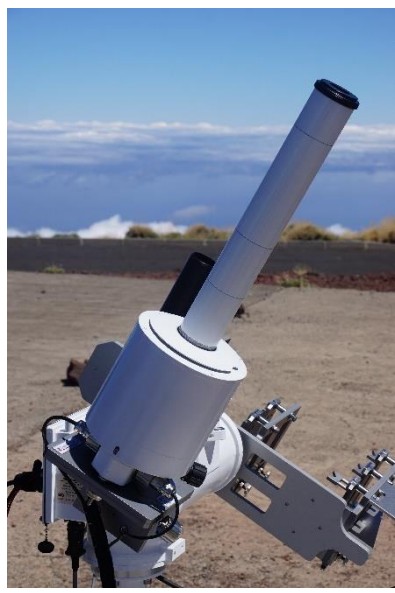

**Figure 3** The BTS2048-UV-S-WP modified for direct solar measurements mounted on a solar tracker at the Izaña observatory in Tenerife.

For outdoor measurements of direct solar irradiance, weather-proof (BTS2048-UV-S-WP) version of the BTS2048-UV-S was constructed. The instrument was integrated in a weather-proof housing which is temperature controlled (ambient temperature range from -25 °C to +50 °C) and water proof. During the three-week measurement campaign in September 2016 at the Izaña observatory (Altitude: 2.390 metres Coordinates: longitude 16º 30´ 35" West, latitude 28º 18´ 00" North), the typical temperature variation within the housing was below 0.1 °C. There was cloudless sky during the measurements

which data has been used for this intercomparison. The housing was equipped with an entrance optic tube to limit the field of view to 2.8° (full opening angle). This tube is based on a baffle design to prevent stray light hitting the diffusor. Mounted on a solar tracker (EKO STR-32G) with a pointing accuracy of < 0.01°, the instrument measured direct solar irradiance. Solar measurements were performed using the solar bandpass correction method. Here, several narrow bandpass filters are used in the spectral range between 280 nm and 420 nm. This allows to measure the steep slope of the solar spectrum below 300 nm

with a high dynamic (see Figure 4). A full spectrum was recorded every 8 seconds.

The measurements were compared to the results of the double-monochromator-based QASUME instrument which is extensively characterised for global irradiance (Gröbner *et al.,* 2005, Gröbner *et al.,* 2005, Hülsen *et al.,* 2016). For this intercomparison the QASUME instrument was equipped with a collimator based entrance optic for direct solar irradiance measurements with a maximum field of view of 2.5 ° (full opening angle, Gröbner *et al.,* 2017). To be able to compare two

sets of data the measured spectra of both instruments had to be synchronised in time and adapted in bandwidth. The QASUME system is operating in sequential mode measuring step by step from lower to higher wavelength. The recording of



a full spectrum from 290 nm to 500 nm in steps of 0.25 nm takes about 16 minutes. Every measurement at a single wavelength is marked with a time stamp so that the corresponding measurement and the wavelength of the BTS2048-UV-S-WP (from now on called BTS) could be synchronised. As both spectroradiometer systems possess different bandwidth, the resulting spectra were convolved with a standard 1 nm triangular bandpass function. This data evaluation results in

5   deviations between BTS and QASUME lower than ± 2.5 % averaged from 300 nm to 420 nm (see Figure 4).

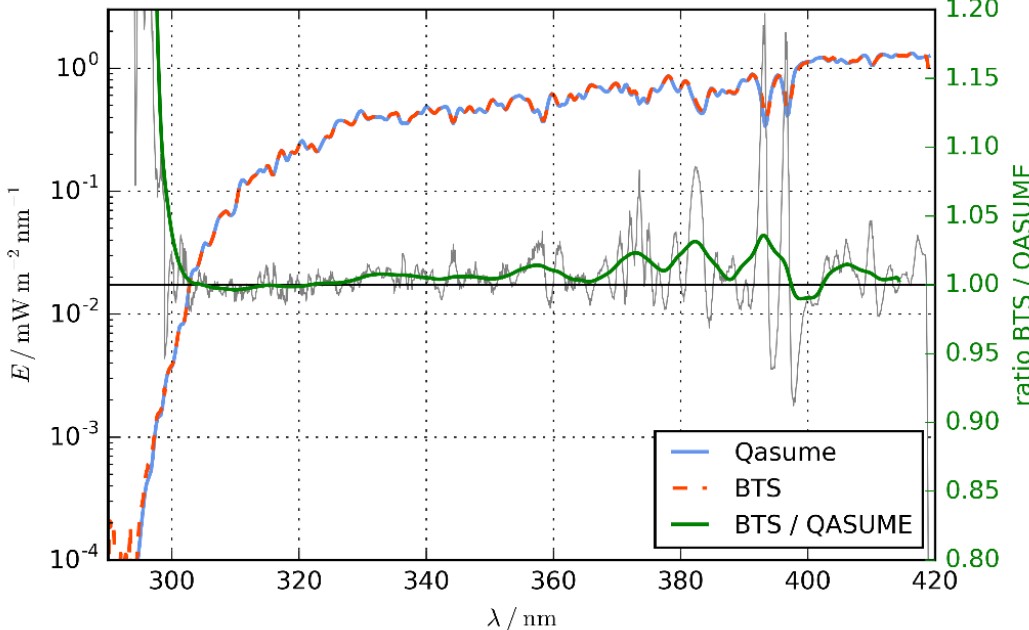

**Figure 4** Direct spectral irradiance (blue and red line) measured in Izana on 20 September, 10:45 UTC by the BTS and the double-monochromator-based QASUME instrument (left axis). The ratio (grey line for single data and green line for moving average) of measurements (right axis) shows satisfactory agreement with average deviations of less than 2.5 % between 300 nm and 420 nm.





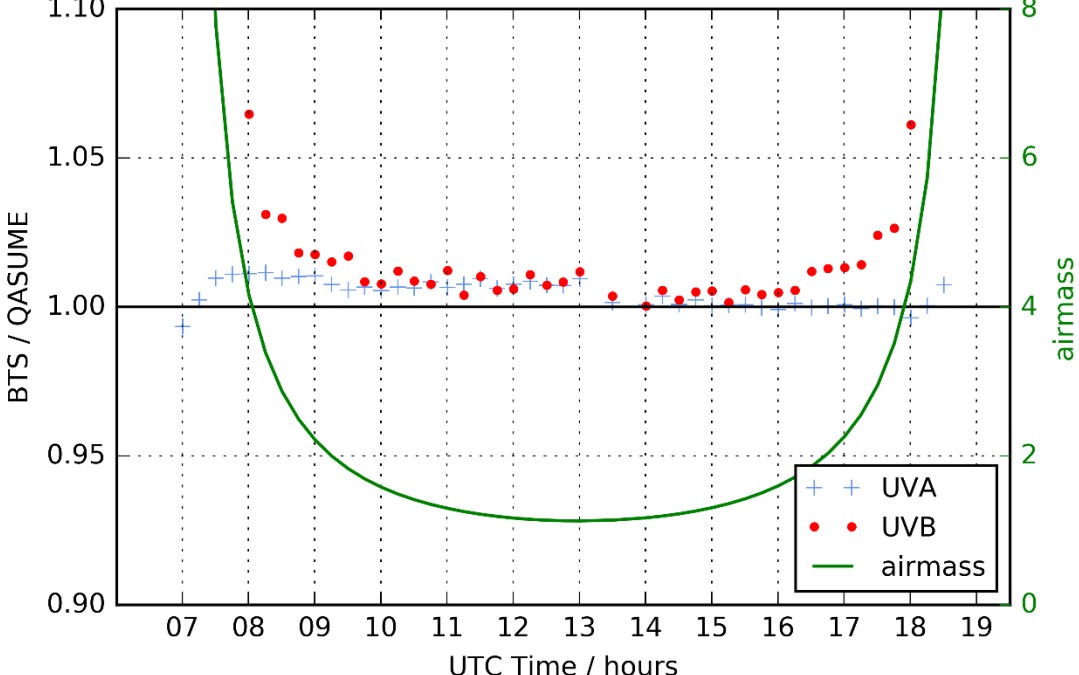

**Figure 5** Ratios of the integrated UV-A (315 nm – 400 nm) and UV-B (280 nm – 315 nm) irradiance between the BTS and the QASUME instrument on 20 September 2016 (left axis) and corresponding airmass (right axis). Only if the airmass is above 6 deviations in the UV-B exceed 5%.

The diurnal variation between QASUME and BTS of the integral UV-A (315 nm to 400 nm) and UV-B (280 nm to 315 nm) irradiance ratios show deviations of less than ± 2 %. The ratio rises for UV-B at an airmass larger than 4 (Figure 5).

**4.2 Intercomparison of TOC values**

The spectral data have also been used to calculate the total ozone column based on a retrieval algorithm proposed by Masserot *et al.* (2002). For this purpose, the ratios of two wavelength bands, ranging from 305 nm to 310 nm and from

340 nm to 350 nm, have been calculated. The ratio is directly related to the TOC since the first band lies inside and the second one outside the ozone absorption range. By comparing these ratios to a set of pre-calculated model values stored in a look-up table, the most probable TOC value present during the measurement can be determined. The model values have been calculated with the libRadtran software package for radiative transfer calculations (Mayer *et al.,* 2012). The look-up table is a data cube with three dimensions and consists of roughly 6500 direct irradiance spectra with a wavelength range of 280 nm

to 420 nm for solar zenith angles (SZA) between 24° and 90° and for TOC values between 250 DU and 350 DU. For the calculation of the ozone value of a specific measurement, all modelled spectra at the SZA apparent during the time of the measurement are first selected from the look-up table (see Figure 6) and the ratios between the BTS measurement and all selected spectra are calculated. For the calculation of the look-up table, the following values for the input parameters have



been chosen: An albedo of 0.2, a pressure of 773 hPa, an altitude of 2.36 km, an atmospheric profile typical for mid-latitude summer (Anderson *et al.,* 1986), and the ozone cross-section of Bass and Paur (1984). The temperature and ozone profile of the chosen atmospheric profile lead to an effective ozone temperature of 232.3 K. In contrast to Masserot et al., direct irradiance instead of global irradiance has been modelled as input for the look-up table to adapt the algorithm to the

5 measurements performed with the BTS. Despite of the low aerosol content in Izaña the aerosol default values of libRadtran have been used. This crude modelling of aerosol parameters is intentional in order to reflect the usually limited knowledge of the atmospheric aerosol properties and therefore shows the robustness of the applied algorithm. The chosen aerosol parameters will lead to deviations between measured and modelled spectra over the whole wavelength range due to differences between the actual atmospheric condition and the assumptions made for the modelled spectra. This is addressed

by performing a linear fit to the ratio between 330 nm and 355 nm (see Figure 7) where ozone absorption is negligible and no "local" spectral features due to, e.g., strong absorption lines in the solar spectrum are apparent. The derived linear fit is applied to each ratio afterwards, effectively adjusting the ratios for atmospheric scattering processes with low wavelength dependencies (e.g. Mie scattering by aerosols or cloud droplets). The ratios are then averaged in the two wavelength bands from 305 nm to 310 nm and 340 nm to 350 nm, respectively. The resulting numbers are divided by each other for each ratio.

The ratio closest to one corresponds to the modelled spectrum with the most likely ozone value apparent during the BTS measurement.

In Figure 8, the results of the TOC calculations based on spectra of the direct solar irradiance measured on the 20 September 2016 are shown. In addition, measurements performed on the same day with other instruments are displayed, namely with the WRC QASUME spectroradiometer, the world primary standard Dobson ozone spectrophotometer D083 from the World

Dobson Calibration Center (WDCC) at NOAA, the regional primary reference Brewer spectrophotometer B157 of the Izaña observatory, and satellite measurements from the Aura ozone monitoring instrument. Additionally, to directly compare the BTS-retrieved TOC to the other instrument data, the mean TOC values have been calculated during the time between 9 and 11 UTC. For the comparison with the Aura Ozone Monitoring Instrument (OMI) data, the BTS data has been averaged ± 15 minutes around the fly-over time of the Aura satellite, resulting in a BTS TOC of 270 DU. The results are illustrated in

Table 1.

| Instrument | TOC / DU | Time interval in UTC | Difference to BTS / % |
|---|---|---|---|
| BTS | 267.6 | 9:00 – 11:00 | - |
| QASUME | 271.7 | 9:00 – 11:00 | +1.5 |
| IZO Brewer | 270.1 | 9:00 – 11:00 | +0.9 |
| NOAA Dobson | 265.5 | 9:00 – 11:00 | -0.8 |
| BTS | 270 | 12:30 – 13:00 | - |
| Aura Ozone Monitoring  (OMI) | 275 | 12:30 – 13:00 | +1.8 |



**Table 1** Comparison of the BTS-retrieved TOC values to other instrument values measured during the ATMOZ campaign. The data has been calculated based on measurements performed on the 20 September 2016 between 9 and 11 UTC, shown in Figure 8. For the comparison to the OMI data the average of the BTS TOC values measured between 12:30 and 13:00 UTC has been calculated.

The systematic differences between BTS and QASUME TOC values, even if the spectra of both instruments agree well as shown in Figure 4, arise from different model approaches which are used for the TOC determination. In addition, the modelled TOC values of BTS and QASUME are based on slightly different input parameters for the atmospheric conditions. This is the case since we have chosen our parameters without knowledge of the QASUME parameters to ensure an unbiased comparison.

During the other measurement days of the campaign, where direct irradiance measurements were performed with the BTS spectroradiometer, the deviation to the other instruments did not exceed 3 % between 9 and 17 UTC. At air masses larger than 4 during sunrise and sunset, the signal to noise ratio decreases in the shortwave region of the spectrum and, therefore, the TOC estimations become noisier. In addition, at lower irradiance levels the detection threshold of the instrument increasingly affects the wavelength band from 305 nm to 310 nm which leads to higher systematic uncertainties for the calculation. A first analysis of the TOC determination uncertainty of the BTS-device which is in the range of 5 DU, has been carried out by Vaskuri et al. (2017) based on a Monte Carlo method.

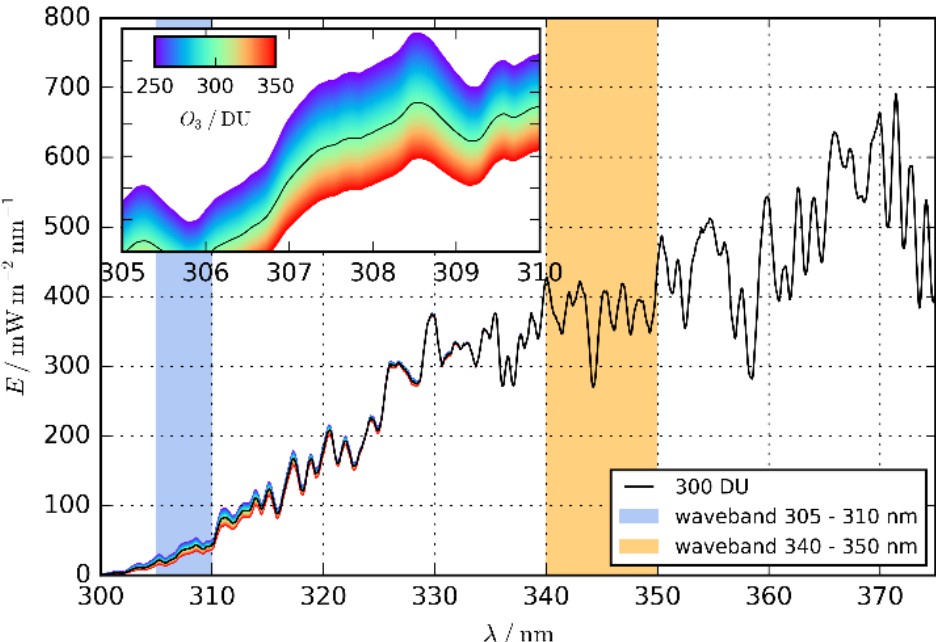

**Figure 6** Direct irradiance spectra of the lookup-table modelled with libRadtran. Shown are all spectra for an SZA of 48°, ranging from 250 DU (purple) to 320 DU (red) TOC. The wavebands used for the TOC retrieval are marked blue and orange, respectively.




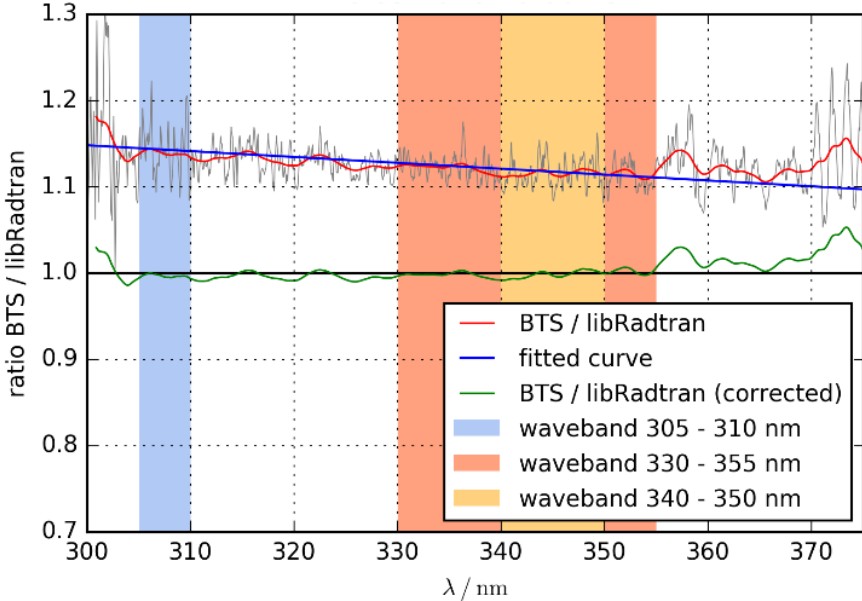

**Figure 7** Example of the atmospheric correction applied to the TOC calculation. The waveband ranging from 330 nm to 355 nm was used to derive the slope of the blue fitting line. The green line is the ratio of a BTS measurement and a libRadtran calculation after subtracting the blue fitting line."

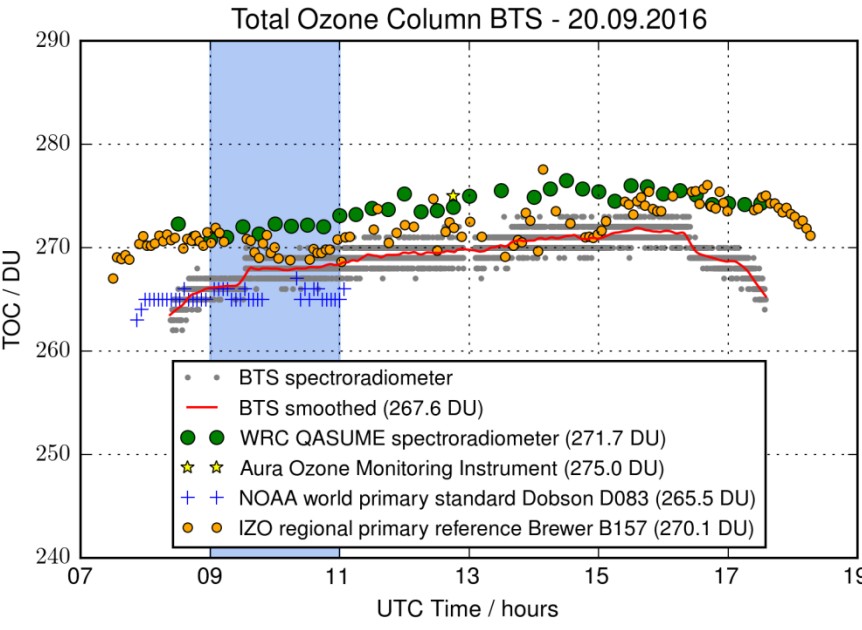

**Figure 8** Total ozone column (TOC) derived by the direct solar measurements of the BTS in comparison to other instruments. The measurements have been conducted during the ATMOZ intercomparison campaign on 20 September 2016.



The TOC values stated in the legend for each ground-based instrument have been derived by averaging the values between 9 and 11 UTC (blue area). Grey dots symbolize the BTS TOC measurements captured every 8 seconds, the digitization steps result from the 1 DU resolution of the look-up table.

## 5. Conclusion

The BTS2048-UV-S series spectroradiometer is a versatile measurement system to perform spectroradiometric measurements in the UV spectral range. Its compact design, the fast sensor system, and the hardware based stray light correction achieved with several optical filters enables a wide range of radiometric applications.

After adapting the BTS2048-UV-S with a weather-proof housing for direct solar irradiance measurements (BTS2048-UV-S-WP) and an extensive device characterisation, the array spectroradiometer proved its capability in the challenging measurements of solar irradiance for atmospheric research.

Absolute direct solar irradiance measurements by the BTS2048-UV-S-WP showed deviations from the double-monochromator-based QASUME lower than ± 2.5 % averaged over the spectral range from 300 nm to 420 nm. TOC values derived from BTS2048-UV-S-WP data show agreement comparable to those between Dobson and Brewer reference instruments.

### Acknowledgement

Many measurements have been carried out in the framework of the EMRP project ENV59 ATMOZ at the Izaña observatory in Tenerife. The authors thank Julian Gröbner from PMOD for providing the QASUME data sets.

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
