# Peer review of "Adaption of an array spectroradiometer for total ozone column retrieval using direct solar irradiance measurements in the UV spectral range"

_Atmospheric Measurement Techniques, 2017_

## Referee Comment (RC1) · Anonymous Referee #1 · 6 Oct 2017

General comments

The paper shows promising results of a newly developed instrument measuring direct irradiance in the UV range to retrieve total ozone columns. There are several interesting features in the instrument, like stray light reduction through sets of filters in the system, seemingly extremely good linearity and radiometric stability. However the paper nearly entirely lacks basic technical description of the covered subjects (instrument description, calibration, algorithm description) and mostly does not even give references (see specific comments 1, 3, 4, 5, 6, 10, 12, 16, 17). For some cases, where references are given, important numbers should be given in the paper itself (see specific comments 9

and 19). Due to the large quantity of missing technical information, I do not consider this paper worth of being published in AMT in its current form. I would recommend the authors to address the comments and resubmit it in a greatly modified version.

Specific comments

1. Section 2 (S2): Give some CCD detector characteristics, e.g. how many pixels does it have, does it have an electronic shutter, or give a reference.

2. S2: "This detector can be spectral mismatch-corrected . . .": does this refer to a comparison of the measured data with the solar Fraunhofer structure as it is mentioned later in the paper? If yes, what does this have to do with the detector? It can always be done.

3. S2: Define the expression "BiTec sensor system", describe it and/or give reference.

4. S2: What set of optical filters is in the system. Exactly the same ones as described in Shaw et al., 2008?

5. S2: Give a reference for the statement that the stray light correction method based on the LSF improves the measurement threshold by 2 orders of magnitude.

6. S2: How long does it take to get one spectra with this instrument?

7. S3: "After an initial settlement . . . ": does "initial settlement refer to temperature stabilization?

8. S3: "After an initial settlement . . . accuracy was found to be better than 0.1 nm": what does this mean? The dispersion could not be determined better than 0.1 nm?

9. S3: " . . . the wavelength scale was adapted with high accuracy . . .": give a value for the accuracy even if it may be listed in the referenced paper Egli et al., 2014.

10. S3: " . . . linearity with a deviation smaller than 1% in the full dynamic range . . .": this is a very small value. There should be an explanation why this is so good. Is this

detector so great or is the "full dynamic range" not starting with very low counts? A figure could be added. Or give a reference.

11. S4.1: " . . . the entrance optic is kept clean with a constantly heated dry airflow.": describe this more in detail or give a reference. How does the air flow inside the entrance optic tube, with which the instrument was equipped for the measurements at Izana?

12. S4.1: Describe the tracker, which was used or give a reference.

13. S4.1: Do you mean 0.01° pointing accuracy or precision or resolution (step)? If you mean accuracy, explain why the accuracy is that good or give a reference. Does it have a quadrant detector or similar system included?

14. S4.1: What is the difference between the gray and green line in figure 4?

15. S4.1: For the (excellent) comparison with Qasume (figure 4): was the radiometric calibration for BTS done in the laboratory and not repeated in the field? In this case the instrument seems to be very robust with respect to transport, which should definitely be mentioned as a strength in the paper.

16. S4.2: The ozone algorithm is based on Masserot et al., which is an algorithm for global measurements (not direct sun data). How was the algorithm modified?

17. S4.2: What effective ozone temperature is used in the algorithm?

18. S4.2: Table 1: the value of 270 DU for OMI seems to be a typo (based on figure 8 it should be around 275 DU).

19. S4.2: "A first analysis of the TOC determination uncertainty . . .": although a reference is given, the authors should still give a number for the uncertainty determined in Vaskuri et al. at this place, so that the reader does not have to look it up himself.

20. S4.2: " . . . At air masses larger than 4 . . . the signal to noise ratio decreases . . . becomes less reliable." Are the data just getting more noisy or are there systematic

effects happening?

21. S4.2: Figure 8: why are there several gray dots at the same time?

22. S4.2: Figure 8: how do the authors explain the decreasing TOC from BTS between 16:00 and 18:00 UT? A residuals stray light effect? An algorithm issue?

23. S4.2: It is somewhat surprising to me that the Qasume and BTS spectra are nearly identical (figure 4), but there is a systematic TOC bias. The authors should elaborate on this, which might need more description of the algorithm.

24. It seems that the Ozone algorithm is purely based on lab-calibration. If this is true, the authors should mention this positively in the paper.

---

## Author Comment (AC1) · 10 Oct 2017

**Author's response amt-2017-240:**

The authors would like to thank the referees for their comments. In the following our response is illustrated.

(1)  Comments from referees/public

1: Section 2: Give some CCD detector characteristics, e.g. how many pixels does it have, does it have an electronic shutter, or give a reference.

2: Section 2: "This detector can be spectral mismatch-corrected ...": does this refer to a comparison of the measured data with the solar Fraunhofer structure as it is mentioned later in the paper? If yes, what does this have to do with the detector? It can always be done.

3: Section 2: Define the expression "BiTec sensor system", describe it and/or give reference.

4: Section 2: What set of optical filters is in the system. Exactly the same ones as described in Shaw et al., 2008?

5: Section 2: Give a reference for the statement that the stray light correction method based on the LSF improves the measurement threshold by 2 orders of magnitude.

6: Section 2: How long does it take to get one spectra with this instrument?

7: Section 3: "After an initial settlement ... ": does "initial settlement refer to temperature stabilization?

8: Section 3: "After an initial settlement ... accuracy was found to be better than 0.1 nm": what does this mean? The dispersion could not be determined better than 0.1 nm?

9: Section 3: " ... the wavelength scale was adapted with high accuracy ...": give a value for the accuracy even if it may be listed in the referenced paper Egli et al., 2014.

10: Section 3: " ... linearity with a deviation smaller than 1% in the full dynamic range ...": this is a very small value. There should be an explanation why this is so good. Is this detector so great or is the "full dynamic range" not starting with very low counts? A figure could be added. Or give a reference.

11: Section 4.1: " ... the entrance optic is kept clean with a constantly heated dry airflow.": describe this more in detail or give a reference. How does the air flow inside the entrance optic tube, with which the instrument was equipped for the measurements at Izana?

12: Section 4.1: Describe the tracker, which was used or give a reference.

13: Section 4.1: Do you mean 0.01° pointing accuracy or precision or resolution (step)? If you mean accuracy, explain why the accuracy is that good or give a reference. Does it have a quadrant detector or similar system included?

14: Section 4.1: What is the difference between the gray and green line in figure 4?

15: Section 4.1: For the (excellent) comparison with Qasume (figure 4): was the radiometric calibration for BTS done in the laboratory and not repeated in the field? In this case the instrument seems to be very robust with respect to transport, which should definitely be mentioned as a strength in the paper.

16: Section 4.2: The ozone algorithm is based on Masserot et al., which is an algorithm for global measurements (not direct sun data). How was the algorithm modified?

17: Section 4.2: What effective ozone temperature is used in the algorithm?

18: Section 4.2: Table 1: the value of 270 DU for OMI seems to be a typo (based on figure 8 it should be around 275 DU).

19: Section 4.2: "A first analysis of the TOC determination uncertainty ...": although a reference is given, the authors should still give a number for the uncertainty determined in Vaskuri et al. at this place, so that the reader does not have to look it up himself.

20: Section 4.2: " ... At air masses larger than 4 ... the signal to noise ratio decreases ... becomes less reliable." Are the data just getting more noisy or are there systematic effects happening?

21: Section 4.2: Figure 8: why are there several gray dots at the same time?

22: Section 4.2: Figure 8: how do the authors explain the decreasing TOC from BTS between 16:00 and 18:00 UT? A residuals stray light effect? An algorithm issue?

23: Section 4.2: It is somewhat surprising to me that the Qasume and BTS spectra are nearly identical (figure 4), but there is a systematic TOC bias. The authors should elaborate on this, which might need more description of the algorithm.

24: It seems that the Ozone algorithm is purely based on lab-calibration. If this is true, the authors should mention this positively in the paper.

(2)  Author's response

1: Section 2: Done
2: Section 2: The spectral mismatch correction is implemented for measurements of conventional UV sources and it is not relevant for solar measurements. The sentence is removed.
3: Section 2: Sentence removed, not relevant in this context.
4: Section 2: We think it is beyond the scope of the paper to provide the technical details of each filter. However we provide basic information about the filters.
5: Section 2: Done
6: Section 2: We agree that this is of interest. However the measurement time depends on the light source, resolution, etc. This is why we stated a typical measurement time.
7: Section 3: Initial settlement is referred to a period of few weeks after the instrument was first used. The sentence was modified to avoid confusion.
8: Section 3: The remaining uncertainty for the overall standard wavelength calibration was 0.1 nm.
9: Section 3: Done
10: Section 3: We totally agree, the sentence had to be modified, as the **remaining** deviation from nonlinearity after correction is smaller than 1%.
11: Section 4.1: This sentence had to be modified as the air flow is not passing the tube and thus is not cleaning the entrance optics.
12: Section 4.1: Done
13: Section 4.1: 0.01° pointing accuracy is achieved by EKO STR-32G trackers. Since we stated the type of sun tracker as a reference it should be clear now.
14: Section 4.1: We modified the sentence to make it clearer.
15: Section 4.1: The radiometric calibration has been carried out in the lab before and after the intercomparison as well as in the field. An additional sentence in section 3 points out the robustness of the instrument as suggested.
16: Section 4.2: A sentence has been added to make it clearer.
17: Section 4.2: A sentence has been added.
18: Section 4.2: Corrected
19: Section 4.2: Done
20: Section 4.2: An explaining sentence has been added.
21: Section 4.2: An explaining sentence has been added.
22: Section 4.2: An explaining sentence has been added (see comment 20).
23: Section 4.2: We agree to the reviewers concern and added a discussion regarding the observed difference between BTS and QASUME TOC.
24: Section 4.2: See comment 15.

(3)  Author's changes

The green text was added, red deleted:

1: Section 2: The spectrometer uses a back-thinned CCD detector with 2048 pixels and an electronic shutter integrated in a compact optical bench.
2: Section 2: This photodiode detector can be spectral mismatch-corrected (CIE, 2016) during the measurement by the spectrometer measurements with regard to its radiometric value.
3: Section 2: Therefore, this advanced BiTec sensor system (BiTec since two technologies are combined, integral photodiode and spectroradiometer) combines the advantages of two sensor technologies to perform reliable measurements in the UV spectral range.
4: Section 2: In the device one yellow glass filter and four interference filters with different wavelength ranges are integrated.
5: Section 2: (Zong et al., 2006)
6: Section 2: Hence the overall measurement time to get a full spectral measurement is the sum of all integration times of the sub measurements. Typically this measurement time is in the range of a few seconds, depended on the light source to measure.
7: Section 3: After an initial settlement of the instrument, the wavelength accuracy The uncertainty for the wavelength calibration was found to be better than 0.1 nm.
8: Section 3: See 7.

9: Section 3: For measurements of solar irradiance however, the wavelength scale was additionally adapted with high accuracy standard deviations better than 0.02 nm to the solar Fraunhofer lines using the MatShic algorithm (Egli et al., 2014).

10: Section 3: By applying a mathematical correction for nonlinearity, the spectrometer showed linearity with a deviation smaller than 1 % over in the full dynamic range for the characterised measurement mode.

11: Section 4.1: The instrument was integrated in a weather-proof housing which is temperature controlled (ambient temperature range from -25 °C to +50 °C) and water proof., and the entrance optic is kept clean with a constantly heated dry airflow.

12: Section 4.1: Mounted on a solar tracker (EKO STR-32G) with a pointing accuracy of < 0.01°, the instrument measured direct solar irradiance.

13: Section 4.1:  See 12.

14: Section 4.1: The ratio (grey line for single data and green line for moving average) of measurements (right axis) shows satisfactory agreement with average deviations of less than 2.5 % between 300 and 420 nm.

15: Section 3: The used standard lamps allow recalibration of the instrument in the laboratory and in the field. During the measurement campaign, described below, the radiometric calibration of the BTS2048-UV-S could be verified with a standard deviation of less than 1% compared to the calibrations in the laboratory before and after the campaign.

16: Section 4.2: In contrast to Masserot et al., direct irradiance instead of global irradiance has been modelled as input for the look-up table to adapt the algorithm to the measurements performed with the BTS.

17: Section 4.2: The temperature and ozone profile of the chosen atmospheric profile lead to an effective ozone temperature of 232.3 K.

18: Section 4.2: Aura Ozone Monitoring (OMI) 270275 TOC/DU

19: Section 4.2: A first analysis of the TOC determination uncertainty of the BTS-device, which is in the range of 5 DU, has been carried out by Vaskuri et al. (2017) based on a Monte Carlo method.

20: Section 4.2: At air masses larger than 4 during sunrise and sunset, the signal to noise ratio decreases in the shortwave region of the spectrum and, therefore, the TOC estimations become less reliable more noisy. In addition, at lower irradiance levels the detection threshold of the instrument increasingly affects the wavelength band from 305 nm to 310 nm which leads to higher systematic uncertainties for the calculation.

21: Section 4.2: Figure 8 Grey dots symbolize the BTS TOC measurements captured every 8 seconds, the digitization steps result from the 1 DU resolution of the look-up table.

22: Section 4.2: See 20.

23: Section 4.2: The systematic differences between BTS and QASUME TOC values, even if the spectra of both instruments agree well as shown in Figure 4, arise from different model approaches which are used for the TOC determination. In addition, the modelled TOC values of BTS and QASUME are based on slightly different input parameters for the atmospheric conditions. This is the case becauses since we have chosen our parameters without knowledge of the QASUME parameters to ensure an unbiased comparison.

24: Section 4.2: See 15.

---

## Short Comment (SC1) · 15 Dec 2017

The units of $mWm^{-2}nm^{-1}$ of Figure 4 are incorrect by a factor of 1000. The units should be $Wm^{-2}nm^{-1}$, which would mean that the detection limit of the system is about $10^{-4}Wm^{-2}nm^{-1}$. State-of-the-art double-monochromator based scanning spectroradiometers such as QASUME have a detection limit of about $10^{-6}Wm^{-2}nm^{-1}$ and sometimes better, in particular for direct solar measurements where no cosine diffuser with small cosine error is required. Such diffusers tend to attenuate radiation more than entrance optics for direct measurements.

The units of Figure 4 should be corrected and the reasons why the detection limit

is about two orders of magnitude worse than that of QASUME should be discussed. Is the difference due to stray light despite the physical stray light suppression of the BTS2048-UV-S system or because of the short integration times of this system and the resulting photon noise? Below 305 nm, the ratio BTS /QASUME shown in Figure 4 is greatly increasing towards shorter wavelength. This is a clear indication of stray light and might be an indication that the stray light suppression of the BTS via the use of interference filters is inferior to that of the double-monochromator based QASUME system. On the other hand, data of the system shown in Figure 5 of the article "Effective stray light suppression with the BTS2048-UV series array spectroradiometer" published in issue 12 of the Thematic Network for Ultraviolet Measurement (http://metrology.tkk.fi/uvnet/source/UVNews_12.pdf) suggests that the BTS has a detection limit of $10^{-5}Wm^{-2}nm^{-1}$ with no obvious sign for stray light, although judging stray light characteristics on a logarithmic scale can be deceptive.

While the accuracy of measurements below 305 are of little relevance for the ozone retrievals described in the paper, a quantitative assessment of the system's stray light characteristics below this wavelength would be of great interest to gauge the potential suitability of the system and its novel physical stray light suppression method for those solar measurement application that were up to this date the domain of scanning double-monochromator instruments.

---

## Short Comment (SC2) · 15 Dec 2017

The anonymous referee had the comment "4: Section 2: What set of optical filters is in the system. Exactly the same ones as described in Shaw et al., 2008?" and the authors replied "We think it is beyond the scope of the paper to provide the technical details of each filter. However we provide basic information about the filters."

I agree with the referee that more information on the filters should be provided and do not feel that enough "basic information about the filters" is included in the manuscript.

Currently, the manuscript only provides the following information:

P2, L15: "In the device one longpass filter and four interference filters with different wavelength ranges are integrated."

**and**

P5, L13: "Here, several narrow bandpass filters are used in the spectral range between 280 nm and 420 nm."

The authors should specify the center wavelength and bandwidth of each narrow bandpass filter (an additional figure with the filters' transmission functions would also be helpful) and describe how the corrected spectrum is obtained from measurements using each of these filters. The novelty of the instrument is its stray light correction scheme and the manuscript should therefore describe how this algorithm works, both in terms of hardware and data processing. I don't consider this "technical details " that are "beyond the scope of the paper".

---

## Referee Comment (RC2) · Anonymous Referee #2 · 16 Dec 2017

Zuber et al., 2017 describe a new spectroradiometer instrument (BTS) for UV measurements between 200 and 430 nm and its application to ozone total column observations. The authors present direct solar irradiance spectral measurements during one day (20 September 2016) at the Izaña Atmospheric Observatory (IZO) and compare them to the QASUMI instrument spectra. The paper also describes total ozone columns (TOC) derived from BTS during the selected day and compares them to TOC from QASUMI, Brewer, Dobson and OMI instruments. The BTS instrument is distinct due to application of different long and short pass filters that are designed to eliminate effects of internal stray light on the accuracy of UV measurements. The topic of the paper is appropriate for publication in ATM. I recommend publishing the paper after appropriate

modifications are made.

Main comments:

1. The authors do not provide enough information in the paper to (1) assess the effectiveness of the filter-based method to eliminate stray light and (2) evaluate signal to noise ratio especially as applied to the direct sun irradiance measurements at different solar zenith angles.

2. The authors compare direct solar irradiance spectra measured by their new BTS instrument and QASUMI. Further they compare TOC derived from BTS and QASUMI. However, the differences in derived TOC cannot be directly related to the differences in the spectral measurements since different TOC retrieval schemes were used. I would strongly recommend applying the same retrieval analysis to both QASUMI and BTS datasets for consistency.

Minor comments:

1. p2. Please describe the spectrometer used in BTS2048-UV-S: focal length, f/#, grating, manufacturer, etc.

2. p2. Please provide more information about the detector (e.g. CCD manufacturer, pixel size, well depth, dark current and typical operational temperature)

3. p2. Please explain how SiC photodiode enables fast time resolved radiometric measurements.

4. p2. Please provide a figure showing long-pass filter and four interference filters transmission curves.

5. p3, What is the practical application of the long-pass OoR correction in the system that has multiple band-pass filters? OoR with a long-pass filter is not very accurate due to non-zero transmission in the wavelength of interest.

6. p3, L. 22-23. Please describe "tunable laser systems" used for dispersion characterization. How many wavelengths were measured, what is the wavelength accuracy of the laser line centers.

7. p3, L26. How was the SLF as a function of wavelength described for convolution with high-resolution absorption cross sections? How stable is SLF as a function of temperature and instrument "motion".

8. p4, L3. Reference for linearity characterization methods is missing.

9. p4, L9. What was non-linearity before the correction was applied?

10. p4, L17. How was absolute radiometric calibration performed in the field?

11. p4, L18. Is 1% stability applicable to all wavelengths?

12. p5 L8. The Izaña Atmospheric Observatory (IZO) altitude is 2373 m a.s.l.

13. p5, L9. Less then 0.1°C is extremely good temperature stability for an outdoor system. Where was temperature sensor located relative to the electronics and spectrometer? What was the set temperature inside the enclosure, what was the outside temperature? What is the temperature controller used? What is the enclosure material?

14. p5, L11. Please show a figure with the instrument field of view measurements (in X and Y direction).

15. p5, L12. Please describe how solar tracking accuracy was measured? Tracking accuracy of better than  $0.01^{\circ}$  is extremely good. How was the tracking actually done to ensure such accuracy?

16. p5 L15. Did the integration time change as a function of solar zenith angle or was it constant at 8 seconds per full spectrum independent of SZA? Figure 8 caption suggests a constant integration time.

17. p6, L5. The reported 1.025 ratio of BTS to QASUMI spectra is mainly applicable to

the 305 -350 nm wavelength range. What causes such large residuals at wavelength larger than 350 nm? Were the spectra aligned relative to each other to correct for potential wavelength changes during the measurements? What was the stability of wavelength scale during the field measurements?

18. p6. Figure 4 shows comparison of QASUMI and BTS spectra for a small AMF. It will be very informative to show similar plots for measurements at SZA 80° and 85°. Please describe sources of noise at high SZA in this part of the paper. Provide full wavelength range: 280 – 430 nm. Evaluating signal below atmospheric cutoff will give a better idea about instrument internal stray light correction.

19. p7, L5. Information content of figure 5 can be significantly improved for the purposes of this paper by using more appropriate for O3 measurements wavelength ranges (A: integrated between 305 and 310, B: integrated between 340-350 nm, C:  $330\pm1$  nm and D:  $355\pm1$  nm). Please specify what AMF is plotted in Fig.5. Is it direct + scattered AMF? If yes, can you please show in addition direct sun AMF. Figure 5 shows different pattern in the morning and afternoon for UV-A irradiance ratios. Is it possible that solar tracking accuracy was azimuth angle dependent on that day? Is this behavior present during measurements on other clear sky days?

20. P8, L6. Application of the pre-set aerosol properties in forward radiative transfer calculations does not provide any evidence of "robustness of the applied algorithm". Please rephrase.

21. P9, L11. In the introduction a statement was made that varying integration times and filters can optimize BTS measurements for different purposes. Taking full spectrum from 200 to 430 nm when only 305-310 and 330-360 nm windows are used and fixing the integration time at 8 sec at all solar zenith angles, does not seem to provide an optimal SNR for direct solar irradiance measurements. Please explain why limitations were put in place not to realize full BTS capabilities, as stated in the introduction, for TOC measurements? Figure 8 also shows that the TOC measurements start degrading

at SZA around 71-74 $^{\circ}$  which potentially suggests that dynamic range of the system is smaller than stated earlier.

Table 1: Please add TOC standard deviation for each measurement averaging period

Comments to figures: Please use brackets to distinguish units from fraction symbol in all figures and table

Figure 1: Please expand the information provided, especially in the sensor system part. Provide instrument dimensions.

Figure 4: Please correct the units on the left axis: Solar direct irradiance [W/m2/nm]. Right axis: irradiance ratio (BTS/QASUME). Legend: capitalize QASUME. Please add: averaged BTS/QASUME.

Figure 5: Left axis: Irradiance ratio (BTS/QASUMI)

Figure 6: Please provide left axis scale for the insert graph. Dobson Unit is not a SI unit, please define DU = 2.69E16 molecules/cm2

Figure 7: Left axis: Irradiance ratio (BTS/LibRadtran) Specify date/time of the measurement

Figure 8: OMI TOC is hard to see. Digitization of the BTS TOC is not entirely clear. Please explain in the text.

---

## Author Comment (AC2) · 11 Jan 2018

**Author's response amt-2017-240:**

The authors thank the referees for their helpful comments. Our responses to all comments are described below.

(1) Comments from referee 2

1. The authors do not provide enough information in the paper to (1) assess the effectiveness of the filter-based method to eliminate stray light and (2) evaluate signal to noise ratio especially as applied to the direct sun irradiance measurements at different solar zenith angles.
2. The authors compare direct solar irradiance spectra measured by their new BTS instrument and QASUME. Further they compare TOC derived from BTS and QASUME. However, the differences in derived TOC cannot be directly related to the differences in the spectral measurements since different TOC retrieval schemes were used. I would strongly recommend applying the same retrieval analysis to both QASUME and BTS datasets for consistency.

Minor comments:
1. p2. Please describe the spectrometer used in BTS2048-UV-S: focal length, f/#, grating, manufacturer, etc.
2. p2. Please provide more information about the detector (e.g. CCD manufacturer, pixel size, well depth, dark current and typical operational temperature)
3. p2. Please explain how SiC photodiode enables fast time resolved radiometric measurements.
4. p2. Please provide a figure showing long-pass filter and four interference filters transmission curves.
5. p3, What is the practical application of the long-pass OoR correction in the system that has multiple band-pass filters? OoR with a long-pass filter is not very accurate due to non-zero transmission in the wavelength of interest.
6. p3, L. 22-23. Please describe "tunable laser systems" used for dispersion characterization. How many wavelengths were measured, what is the wavelength accuracy of the laser line centers.
7. p3, L26. How was the SLF as a function of wavelength described for convolution with high-resolution absorption cross sections? How stable is SLF as a function of temperature and instrument "motion".
8. p4, L3. Reference for linearity characterization methods is missing.
9. p4, L9. What was non-linearity before the correction was applied?
10. p4, L17. How was absolute radiometric calibration performed in the field?
11. p4, L18. Is 1% stability applicable to all wavelengths?
12. p5 L8. The Izaña Atmospheric Observatory (IZO) altitude is 2373 m a.s.l.
13. p5, L9. Less then 0.1°C is extremely good temperature stability for an outdoor system. Where was temperature sensor located relative to the electronics and spectrometer? What was the set temperature inside the enclosure, what was the outside temperature? What is the temperature controller used? What is the enclosure material?
14. p5, L11. Please show a figure with the instrument field of view measurements (in X and Y direction).
15. p5, L12. Please describe how solar tracking accuracy was measured? Tracking accuracy of better than 0.01° is extremely good. How was the tracking actually done to ensure such accuracy?
16. p5 L15. Did the integration time change as a function of solar zenith angle or was it constant at 8 seconds per full spectrum independent of SZA? Figure 8 caption suggests a constant integration time.
17. p6, L5. The reported 1.025 ratio of BTS to QASUMI spectra is mainly applicable to the 305 -350 nm wavelength range. What causes such large residuals at wavelength larger than 350 nm? Were the spectra aligned relative to each other to correct for potential wavelength changes during the measurements? What was the stability of wavelength scale during the field measurements?
18. p6. Figure 4 shows comparison of QASUMI and BTS spectra for a small AMF. It will be very informative to show similar plots for measurements at SZA 80° and 85°. Please describe sources of noise at high SZA in this part of the paper. Provide full wavelength range: 280 − 430 nm. Evaluating signal below atmospheric cutoff will give a better idea about instrument internal stray light correction.
19. p7, L5. Information content of figure 5 can be significantly improved for the purposes of this paper by using more appropriate for O3 measurements wavelength ranges (A: integrated between 305 and 310, B: integrated between 340-350 nm, C: 330+/-1 nm and D: 355+/-1 nm). Please specify what AMF is plotted in Fig.5. Is it direct + scattered AMF? If yes, can you please show in addition direct sun AMF. Figure 5 shows different pattern in the morning and afternoon for UV-A irradiance ratios. Is it possible that solar tracking accuracy was azimuth angle dependent on that day? Is this behavior present during measurements on other clear sky days?
20. P8, L6. Application of the pre-set aerosol properties in forward radiative transfer calculations does not provide any evidence of "robustness of the applied algorithm". Please rephrase.

21. P9, L11. In the introduction a statement was made that varying integration times and filters can optimize BTS measurements for different purposes. Taking full spectrum from 200 to 430 nm when only 305-310 and 330-360 nm windows are used and fixing the integration time at 8 sec at all solar zenith angles, does not seem to provide an optimal SNR for direct solar irradiance measurements. Please explain why limitations were put in place not to realize full BTS capabilities, as stated in the introduction, for TOC measurements? Figure 8 also shows that the TOC measurements start degrading at SZA around 71-74° which potentially suggests that dynamic range of the system is smaller than stated earlier.

Table 1: Please add TOC standard deviation for each measurement averaging period. Comments to figures: Please use brackets to distinguish units from fraction symbol in all figures and table.
Figure 1: Please expand the information provided, especially in the sensor system part. Provide instrument dimensions.
Figure 4: Please correct the units on the left axis: Solar direct irradiance [W/m2/nm]. Right axis: irradiance ratio (BTS/QASUME). Legend: capitalize QASUME. Please add: averaged BTS/QASUME.
Figure 5: Left axis: Irradiance ratio (BTS/QASUMI)
Figure 6: Please provide left axis scale for the insert graph. Dobson Unit is not a SI unit, please define DU = 2.69E16 molecules/cm2
Figure 7: Left axis: Irradiance ratio (BTS/LibRadtran) Specify date/time of the measurement
Figure 8: OMI TOC is hard to see. Digitization of the BTS TOC is not entirely clear. Please explain in the text.

(2)   Author's response
Comments:
General: We appreciate the detailed comments of the referee. The overall goal of the article is a feasibility study rather than a detailed technical report on the various components of the instrument. Although justified by general curiosity, not all examinations requested by the reviewer have been performed. Therefore the focus of our manuscript is on the application and characterization of the BTS device to direct solar irradiance measurements and not in technical engineering details of the meter itself.

1: We think that the effectiveness of the filter-based stray-light elimination has been demonstrated by the comparison to spectral measurements of QASUME instrument which is illustrated in the logarithmic plot. We agree that a plot at a different SZA is beneficial, hence we added a figure showing spectra taken at different SZA to demonstrate the effectiveness of the method and the limitations of the instrument.
2: We agree, that the application of the TOC algorithm to QASUME data would grant a better comparability if we would introduce the same algorithms for both instruments. However, in practice this is difficult or impossible to achieve due to the necessary scanning of wavelengths of the QASUME instrument. A highly sophisticated lookup table with a wavelength per wavelength retrieval of corresponding SZA values would be necessary and its development is beyond the scope of this paper which focuses on the BTS instrument. Our intention is to show that a measurement system which is based on a stock UV spectroradiometer adapted and characterized for direct solar irradiance measurements can determine with the introduced TOC evaluation algorithm comparable results to other systems which use an established TOC algorithm. Another publication is planned about this measurement campaign with all the measurement devices which took part in the intercomparison.

Minor Comments:
1: The authors think it is beyond the scope of this manuscript to present all the details of every single optical part of the spectrometer unit. The important features have been already described and we added even more details.
2: We added the manufacturer and the operational temperature (see section (3) author´s changes).
3: This section is removed, it is not important for the paper.
4: The measurement could be achieved by a different set of optical filters and there is no need to use exactly these filters. A sentence has been added to address this fact (see section (3) author´s changes).
5: The OoR correction method using the long-pass filter is mainly an alternative to the solar bandpass-filter method. It could be used for faster measurements. But due to the limited accuracy it has not been used for the results presented in the paper.
6: The tuneable laser system and its wavelength accuracy is described in the cited Nevas et. Al., 2014 publication and was not part of the feasibility study. The measurements took place every 5 nm over the entire spectral range of the instrument. Figure 2 demonstrates that no significant change of the bandpass function could be observed.
7: For the convolution, a symmetrical bandpass of 0.6 nm was assumed for the spectral range of interest. The temperature and temporal dependence of the bandpass has not yet been investigated. For this paper we assumed that within the temperature change below 0.1 °C no significant change in this will be observable.
8: Reference added (see section (3) author´s changes).
9: Before the linearization the linearity was sufficient for typical measurements, however for low saturation (below 500 cts) the nonlinearity rises above 1% to about 3-5 % for single pixels. This is why a linearization was applied. We added more information (see section (3) author´s changes).
10: For calibration checks, an optical bench with standard lamps was installed in the Izana Labs. The instrument was removed from the tracker and recalibrated in the Lab. We added a statement (see section (3) author´s changes).
11: For the spectral range of solar measurements used in this paper (280 nm to 420 nm),  yes.
12: Corrected following observatory website
http://izana.aemet.es/index.php?option=com_content&view=article&id=21&Itemid=21&lang=en
13: Additional information on the temperature sensor will be added to the paper (see section (3) author´s changes). Further information on controller and enclosure material is property of the instrument manufacturer.
14: Figure 3 has been extended with this data.
15: The tracking accuracy of 0.01° is stated by the company EKO for their stock product which is described in the paper. They use a four quadrant diode for precise fine tracking. Further details have not been investigated, but even if the tracking uncertainty would be higher, it would not be relevant due to the size of solar disc (0.5°) and the FOV of 2.8°.

16. We stated that the measurement interval is given with 8s. The integration time changed within these 8 seconds for the different bandpass filters and for different SZA. We added a sentence (see section (3) author´s changes).

17: This change can be explained with slightly different optical bandpasses of the instruments even after applying a convolution. Since the sun exhibits many Fraunhofer lines this is a typical ratio in such intercomparisons. A sentence has been added (see section (3) author´s changes).

18: We agree and added a second plot by a different SZA. In the spectral range below the solar edge in the UV-B, noise of the 16 bit ADC is dominant and not stray light. This is due to no averaging of the BTS data was used (this was not possible within 8 seconds measurement interval) and the high dynamic of a solar measurement. We wanted to show that even under these circumstances a sophisticated spectroradiometer can perform well for TOC determination. Changes see section (3) author´s changes.

19: We changed the content of Figure 5 accordingly. We specified the figure caption to clarify the airmass given in the figure.

The step of up to 0.5 % occurring around noon indeed is apparent at other days too, but is always connected to the time after QASUME was shortly taken off operation to check its calibration.

20: We removed the statement, since our intention why no special emphasis is given for the aerosol modelling is explained sufficiently in the sentences following that statement.

21: The integration time was adapted for each measurement with each filter (see comment 16). Although the TOC evaluation model only uses the spectral bands between 305 nm to 310 nm and 340 nm to 350 nm, the whole solar UV spectrum was measured. This allowed the spectral comparison to QASUME. We assume that future TOC evaluation models can utilize further parts of the solar spectrum and we could already demonstrate the capability of spectral measurements. Due to a measurement interval of 8 seconds no averaging or very long integration times were performed which limits the BTS capabilities at high SZA.

Table 1: The TOC uncertainty evaluation was evaluated in the paper of Vaskuri 2017 which is cited. The representation of the units on the x and y axis is following the ISO 80000-1:2009 specifications.

Figure 1: More information is provided (see section (3) author´s changes).

Figure 4: We corrected the left axis according to the recommendation of the reviewer to "solar direct irradiance / W/m2/nm". The right axis was corrected to "irradiance ratio (BTS / QASUME)". We capitalized the "QASUME" in the legend and added "averaged BTS/QASUME".

Figure 5: We adapted the figure as recommended.

Figure 6: We adapted the graph and introduced the DU unit definition in the manuscript.

Figure 7: We added date/time of the measurement

Figure 8: A sentence has been added (see section (3) author´s changes).

(3)  Author's changes

The green text was added, red deleted:

1: Additional figure at different SZA added.
2: -

Minor Comments:
1: -
2: Sentence modified: The spectrometer uses a temperature controlled (8 °C) back-thinned Hamamatsu CCD detector with 2048 pixels and an electronic shutter integrated in a compact optical bench with 16 bit analogue digital converter (ADC) resolution.
3: -
4: Sentence modified: In the device one longpass filter, a bandpass filter (298 nm to 390 nm) and four interference filters (center wavelength: 254 nm, 285 nm, 300 nm and 400 nm) are integrated.
5: -
6: -
7: -
8: Reference added: Tomi Pulli, Saulius Nevas, Omar El Gawhary, Steven van den Berg, Janne Askola, Petri Kärhä, Farshid Manoocheri, and Erkki Ikonen, "Nonlinearity characterization of array spectroradiometers for the solar UV measurements," Appl. Opt. 56, 3077-3086 (2017)
9: Sentence added: "For spectral measurements the instrument saturation is kept below 80% to operate the instrument in an optimum between saturation and linearity. By using the solar bandpass-filter method, with adapted spectral integration times for each sub-measurement, the dynamic range of the instrument could be extended. Whenever the nonlinearity exceeded 1% (at very low signals) a nonlinearity correction is automatically applied.
10: Sentence added: "During the campaign, the instrument was removed from the tracker at night to perform calibration measurements on a portable optical bench."
11: -
12: 2390 m → 2373 m
13: Sentence added "(measured close to the spectrometer unit, set temperature 38°C)."
14: Figure 3 edited and additional reference added in the description.
15: -
16: Senctence added "The duration of this measurement interval is mainly accused by filter movement and dark-signal measurements, whereas the integration time for the measurements with different bandpass filters was optimized and varied with different solar zenith angles (SZA)."
17: Sentence added "The higher frequent ratio (grey line) can be explained due to a slightly different optical bandwidth even after the convolution to 1 nm triangular bandpass between BTS/QASUME."
18: Sentence added "At higher SZA and lower input signal the integration time and the noise of the CCD array are increasing which is reducing the signal to noise ratio."
19: -
20: Sentence deleted "and therefore shows the robustness of the applied algorithm"
21: -
Table 1: Column standard deviation added.
Figure 1: Figure provides now more information.
Figure 4: We corrected the left axis according to the recommendation of the reviewer to "solar direct irradiance / W/m$^{-2}$·nm$^{-1}$". The right axis was corrected to "irradiance ratio (BTS/QASUME)".
We capitalized "QASUME" in the legend and added "averaged BTS/QASUME".
Figure 5: We adapted the figure as recommended.
Figure 6: We adapted the graph and introduced the DU unit definition in the manuscript. (one Dobson Unit (DU) is equivalent to 0.4462 mmol·m$^{-2}$ (Basher, 1982)).
Figure 7: We added date/time of the measurement
Figure 8: A sentence has been added.

(4)   Further Author's changes

In addition the abstract has been slightly rephrased to address the comments:

**Abstract.** A compact array spectroradiometer that enables precise and robust measurements of solar UV spectral direct irradiance is presented. We show that this instrument can retrieve total ozone column (TOC) accurately. The internal stray light, which is often the limiting factor for measurements in the UV spectral range and increases the uncertainty for TOC analysis, is physically reduced so that no other stray light reduction methods, such as mathematical corrections, are necessary. The instrument has been extensively characterised at the Physikalisch-Technische Bundesanstalt (PTB) in Germany. During an international total ozone measurement intercomparison at the Izaña observatory in Tenerife, the high-quality applicability of the instrument was verified with measurements of the direct solar irradiance and subsequent TOC evaluations based on the spectral data measured between 12 and 30 September 2016. The results showed deviations of the TOC of less than 1.5 % to most other instruments in most situations not exceeding 3 % compared to established TOC measurement systems such as Dobson or Brewer.

1.   Sentence added in the discussion & results section: "In the spectral range below the UV-B solar edge the deviation rises mostly due to slight differences in the wavelength calibration and insufficient signal to noise ratio since no averaging of the BTS data or longer integration time was possible for the 8 s measurement interval."
2.   Sentence added in the discussion & results section: "Based on the results and the experience gained during the measurement campaign the design and the sensitivity of the BTS measurement systems could be further improved by a factor of 4. This will very likely improve the performance of the system especially at higher SZAs and shall be tested at future measurement campaigns."
3.   Sentence rephrased in the Acknowledgement section: "This work has been supported by the European Metrology Research Programme (EMRP) within the joint research project EMRP ENV59 ATMOZ "Traceability for atmospheric total column ozone". The EMRP is jointly funded by the EMRP participating countries within EURAMET and the European Union. Many measurements have been carried out in the framework of the EMRP project ENV59 ATMOZ at the Izaña observatory in Tenerife."

---

## Author Comment (AC3) · 11 Jan 2018

**Author's response amt-2017-240:**

The authors would like to thank the community member for the comments. In the following our response is illustrated.

(1) Comments from community member

1. The units of mWm-2nm-1 of Figure 4 are incorrect by a factor of 1000. The units should be Wm-2nm-1, which would mean that the detection limit of the system is about 10-4Wm-2nm-1.

2. State-of-the-art double-monochromator based scanning spectroradiometers such as QASUME have a detection limit of about 10-6Wm-2nm-1 and sometimes better, in particular for direct solar measurements where no cosine diffuser with small cosine error is required. Such diffusers tend to attenuate radiation more than entrance optics for direct measurements.

3. The units of Figure 4 should be corrected and the reasons why the detection limit is about two orders of magnitude worse than that of QASUME should be discussed. Is the difference due to stray light despite the physical stray light suppression of the BTS2048-UV-S system or because of the short integration times of this system and the resulting photon noise? Below 305 nm, the ratio BTS /QASUME shown in Figure 4 is greatly increasing towards shorter wavelength. This is a clear indication of stray light and might be an indication that the stray light suppression of the BTS via the use of interference filters is inferior to that of the double-monochromator based QASUME system. On the other hand, data of the system shown in Figure 5 of the article "Effective stray light suppression with the BTS2048-UV series array spectroradiometer"published in issue 12 of the Thematic Network for Ultraviolet Measurement http://metrology.tkk.fi/uvnet/source/UVNews_12.pdf) suggests that the BTS has a detection limit of 10-5Wm-2nm-1 with no obvious sign for stray light, although judging stray light characteristics on a logarithmic scale can be deceptive. While the accuracy of measurements below 305 are of little relevance for the ozone retrievals described in the paper, a quantitative assessment of the system's stray light characteristics below this wavelength would be of great interest to gauge the potential suitability of the system and its novel physical stray light suppression method for those solar measurement applications that were up to this date the domain of scanning double monochromator instruments.

4. The anonymous referee had the comment "4: Section 2: What set of optical filters is in the system. Exactly the same ones as described in Shaw et al., 2008?" and the authors replied "We think it is beyond the scope of the paper to provide the technical details of each filter. However, we provide basic information about the filters." I agree with the referee that more information on the filters should be provided and do not feel that enough "basic information about the filters" is included in the manuscript. Currently, the manuscript only provides the following information: P2, L15: "In the device one long-pass filter and four interference filters with different wavelength ranges are integrated." And P5, L13: "Here, several narrow bandpass filters are used in the spectral range between 280 nm and 420 nm." The authors should specify the center wavelength and bandwidth of each narrow bandpass filter (an additional figure with the filters' transmission functions would also be helpful) and describe how the corrected spectrum is obtained from measurements using each of these filters. The novelty of the instrument is its stray light correction scheme and the manuscript should therefore describe how this algorithm works, both in terms of hardware and data processing. I don't consider this "technical details " that are "beyond the scope of the paper".

(2) Author's response

1: We corrected the axis label accordingly.
2: We agree.
3: Due to the design of the adapted BTS entrance optics, the detection limit was not optimized for direct solar measurements. A small diffusor with comparably high attenuation was used. After the campaign, the instrument sensitivity could be improved by a factor of 4.
The comparison of spectral measurements performed with QASUME and BTS is very complex as both instruments have been operated with different bandpass, wavelength stepping, measurement times and intervals and wavelength accuracy. Especially in the spectral region below 300 nm, where the solar irradiance increases rapidly with decreasing wavelength, the data are more uncertain. So, the differences between the two instruments are not only due to stray light suppression. Nevertheless, the potential and the limitation of the instrument for solar measurements could be demonstrated.
Yes, for different configurations, the detection limit of the BTS is different. The article in UVNews 12 showed results that have been taken with longer measurement times and for global solar irradiance

measurements, where the signal is in general higher and therefore more averaging can be performed. The improved BTS for direct solar measurements will have a similar performance. We added statements which emphasize that a measurement interval of 8 seconds was used; hence no averaging or longer integration times were applied.

4. The focus was on the application and characterization of the BTS device to direct solar irradiance measurements and not in focusing on technical engineering details of the meter itself. It has to be mentioned, that for other applications, the measurement device can be equipped with different sets of interference filters and there is no need to use exactly these filters. Nevertheless, more information about the filters is now given in the manuscript: "In the device one longpass filter, a bandpass filter (298 nm to 390 nm) and four interference filters (center wavelength: 254 nm, 285 nm, 300 nm and 400 nm) are integrated."